

# Freshwater sponges in the southeastern U.S. harbor unique microbiomes that are influenced by host and environmental factors

Jacqueline G. Keleher[1], Taylor A. Strope[1,2], Noah E. Estrada[1], Allison M. Griggs Mathis[1], Cole G. Easson[3] and Cara Fiore[1]

[1] Biology Department, Appalachian State University, Boone, NC, United States
[2] Department of Biochemistry and Molecular Biology, University of Kansas Medical Center, Kansas City, KS, United States
[3] Biology Department, Middle Tennessee State University, Murfreesboro, TN, United States

Corresponding author
Cara Fiore, fiorec@appstate.edu

## ABSTRACT

Marine, and more recently, freshwater sponges are known to harbor unique microbial symbiotic communities relative to the surrounding water; however, our understanding of the microbial ecology and diversity of freshwater sponges is vastly limited compared to those of marine sponges. Here we analyzed the microbiomes of three freshwater sponge species: *Radiospongilla crateriformis*, *Eunapius fragilis*, and *Trochospongilla horrida*, across four sites in western North Carolina, U.S.A. Our results support recent work indicating that freshwater sponges indeed harbor a distinct microbiome composition compared to the surrounding water and that these varied across sampling site indicating both environmental and host factors in shaping this distinct community. We also sampled sponges at one site over 3 months and observed that divergence in the microbial community between sponge and water occurs at least several weeks after sponges emerge for the growing season and that sponges maintain a distinct community from the water as the sponge tissue degrades. Bacterial taxa within the Gammproteobacteria, Alphproteobacteria, Bacteroidota (Flavobacteriia in particular), and Verrucomicrobia, were notable as enriched in the sponge relative to the surrounding water across sponge individuals with diverging microbial communities from the water. These results add novel information on the assembly and maintenance of microbial communities in an ancient metazoan host and is one of few published studies on freshwater sponge microbial symbiont communities.

# INTRODUCTION

The microbial ecology of freshwater sponges is still in the early stages of characterization and discovery relative to marine sponge counterparts. However, interests largely in natural product potential and in the unique high biomass and diversity of sponges in Lake Baikal have driven initial explorations of freshwater sponge microbial
communities (*e.g.*, *Parfenova et al., 2008*; *Kaluzhnaya, Krivich & Itskovich, 2012*; *Costa et al., 2013*; *Keller-Costa et al., 2014*; *Gaikwad, Shouche & Gade, 2016*; *Kaluzhnaya, Lipko & Itskovich, 2021*; *Clark et al., 2022*). This work led to the discovery that freshwater sponges, like their marine counterparts, exhibit some selection of their microbial community, despite the limited abundance of prokaryotic cells in the sponge mesohyl (*Gernert et al., 2005*; *Costa et al., 2013*; *Gaikwad, Shouche & Gade, 2016*; *Sugden et al., 2022*).

One early study of the microbiome associated with a freshwater sponge was that of *Spongilla lacustris* from a lake in Germany, which revealed low microbial abundance and an overall community similar to that of the surrounding water, but also included two deeply branching lineages of Alphaproteobacteria unique to the sponge (*Gernert et al., 2005*). Since the restriction fragment length polymorphism (RFLP) analysis by *Gernert et al. (2005)*, the microbiomes of several additional sponges have been analyzed using low and high throughput sequencing techniques. The sponge species include *Lubomirskia baicalensis* and *Baikalospongia* sp. from Lake Baikal (*e.g.*, *Kaluzhnaya, Krivich & Itskovich, 2012*; *Gladkikh et al., 2014*), *Eunapius carteri* and *Corvospongilla lapidosa* from two lakes in India (*Gaikwad, Shouche & Gade, 2016*), *Ephydatia fluviatilis* in Europe (*e.g.*, *Costa et al., 2013*), *Ephydatia muelleri* in North America (*e.g.*, *Sugden et al., 2022*), *Eunapius fragilis* in North America (*Clark et al., 2022*, and *Tubella variabilis* in Brazil *Laport, Pinheiro & da Costa Rachid, 2019*). Across these studies, the Alphaproteobacteria continue to be prominent within sponge microbiomes, as well as Betaproteobacteria, Cyanobacteria, Actinobacteria, and Verrucomicrobia, while additional groups such as Nitrospira, Chloroflexi, Firmicutes (Bacillota), Planctomycetes, and TM7 among others, may be more variable driven by host species specificity or the local environment (*e.g.*, *Gaikwad, Shouche & Gade, 2016*; *Kohn et al., 2020*; *Sugden et al., 2022*). Interestingly, several studies documented higher diversity of freshwater sponge microbiomes compared to marine microbiomes (*Gaikwad, Shouche & Gade, 2016*; *Laport, Pinheiro & da Costa Rachid, 2019*; *Sugden et al., 2022*); however, low taxonomic diversity has also been documented (*Gernert et al., 2005*). The microbial community of freshwater sponges so far appears to contain few archaea (*Sugden et al., 2022*; *Hustus et al., 2023*) in comparison to marine sponges, but does include microbial eukaryotes, which may include the parasitic protozoa *Cryptosporidium* (*Masangkay et al., 2020*) as well as symbiotic single-celled algae (*Hustus et al., 2023*).

There is not yet a large enough database of freshwater sponge and corresponding water or sediment microbial sequences that would allow us to draw general conclusions about the colonization process of these sponge microbial communities, but so far, the data support a mix of environmental and host influence on the community composition. The extent to which each of these types of factors (environmental *vs.* host) influence freshwater sponge microbiomes and how this might vary across species is not yet known. However, *Laport, Pinheiro & da Costa Rachid (2019)* found that the microbiomes of geographically distant freshwater sponges were still more similar to each other than to those of marine sponges, suggesting salinity as one important environmental factor. Bleaching and mortality, potentially driven by temperature or chemical pollution stress, have also been documented in sponges in Lake Baikal. Specifically, sponges with a decrease in specificity

of microbial taxa (*e.g.*, more common freshwater or sediment taxa), indicated susceptibility of freshwater sponge microbiomes to environmental stress with detrimental impacts to the host (*Kaluzhnaya & Itskovich, 2015*; *Belikov, Petrushin & Chernogor, 2022*).

For many freshwater sponges, an annual life history that includes asexual reproduction through the dormant bodies of gemmules, means there could be temporal host-microbe interactions that choreograph with life stage. This form of asexual reproduction would also suggest that if there is vertical transmission of symbiont lineages, that this transmission must occur *via* gemmules in addition to gametes and/or larvae. While a comparison between gemmule-hatched sponge microbiomes and those of adult sponge tissue from the same river did not support specific vertical transmission of symbiont lineages through gemmule production (*Kenny et al., 2020*; *Sugden et al., 2022*), an experimental analysis with a different sponge species revealed a subset of the microbial community is transmitted vertically on the outer surface an in the interior of gemmules (*Paix, van der Valk & de Voogd, 2024*). There is limited literature on this topic and additional experimental work is needed. While the presence of unique or deeply branching lineages have been documented in freshwater sponges (*Gernert et al., 2005*; *Kaluzhnaya, Krivich & Itskovich, 2012*), there also appears to be year to year and individual to individual variability that suggests more flexible associations between sponges and symbionts allowing for a stronger environmental influence (*Keller-Costa et al., 2014*; *Laport, Pinheiro & da Costa Rachid, 2019*; *Sugden et al., 2022*). Interactions between the sponge host and microbial symbionts likely include the use of eukaryotic-like domains, similar to marine sponges and other systems (*e.g.*, *Díez-Vives et al., 2017* and references therein), but recent work has shown that there may be differences in metabolic traits of freshwater sponge symbionts compared to marine counterparts (*Sugden et al., 2022*). Additional DNA and RNA sequencing of freshwater sponge microbiomes will be needed to better understand the ecology and evolution of these ancient associations. However, for most freshwater sponge species, taxonomic profiling remains a necessary first step toward better characterizing the microbial communities of freshwater sponges.

Here we analyzed the microbiomes of three freshwater sponge species from four sites in western North Carolina, USA. Our goal was to compare the microbiomes of sponges to that of the surrounding water and compare the microbiomes of each sample type over time and across locations. We hypothesized that the microbial communities of the sponges would be distinct from that of the water and that the communities would differ by site and by host species. At one site, sponge and water samples were taken each month during the sponge life cycle from vegetative growth in July, to gemmule formation and degradation of adult tissue by the middle of September, capturing a temporal picture of the sponge microbiome. With this temporal dataset, we hypothesized that the microbiomes of the sponges would be distinct from the water during vegetative growth of the sponge and would look more like the water microbial community as the sponge degraded. Despite an imperfect sampling scheme with locations added over time, these data provide baseline information on the microbial diversity of additional freshwater sponge species and add novel insights on the temporal microbial ecology of these systems.

## MATERIALS AND METHODS

### Site description, sample collection, and nutrient analysis

This dataset began with pilot 16S rRNA gene cloning data from the first year of sponge collections (2016) in this region (Supplemental Information), which motivated a larger geographic search for sponges the following year (2017) and a shift to high throughput sequencing. Because of this, three additional sampling sites were added throughout the year yielding an uneven sampling design and different timing of collection at sites. Sponges and corresponding water samples were collected from a total of four locations near Boone, North Carolina, USA in 2017 (Fig. 1; Table 1): New River at the New River State Park in July, August, and twice in September (NR; collection permit 2021_0548), Watauga River in September (WR), Jacob Fork River in November (JF), and a mussel hatchery pond and indoor tank with mussels in September (Hatch). The hatchery is at a similar elevation to the Jacob Fork River site and while not directly connected, they are in the same water basin (Catawba River basin). In contrast, the New River and Watauga River sites are each in different water basins (New and Watauga River basins, respectively).

All sites were surrounded by a mix of agriculture and woodlands and generally close to a road, which means that all sites were subject to sporadic road runoff as well as agricultural runoff. The largest physical or geographic difference between the sites is elevation as described above and the corresponding air temperature which tends to be several degrees warmer at the lower elevation sites. Sponges in the New, Watauga, and Jacob Fork rivers were found on the underside or sides of rocks that were large enough to not tumble easily with the current. Sponges were never found on small rocks, pebbles, or wood debris at these sites. The rivers of New, Watauga, and Jacob Fork are all fourth order streams and stream flow rate and depth vary considerably over time. At the times of sampling, the water depth generally varied between 0.25 and 1 m depth. After rain events the water frequently rises to 1.5–2 m depth at these sites and the flow rate is generally increased, while after periods of drought the water may be barely covering the rock substrate. The mussel hatchery pond is seasonal and spring fed from the surrounding Catawba River watershed. The sponges collected from the pond were from approximately 0.75 m depth growing on bricks used to weight the mussel aquaculture containers. The catchment areas, calculated by the United States Geological Survey (streamstats.usgs.gov/ss/) for each sampling site are: 606.1 km$^2$ (New River), 244.8 km$^2$ (Watauga), 213.4 km$^2$ (Jacob Fork), and 1.4 km$^2$ (mussel hatchery).

To address the hypotheses that sponge microbiomes would differ 1) from the surrounding water, and 2) across study locations, sponge and water samples were collected at the same time each time of sampling (except for early-September at the New River; Table 1). Individual sponges were photographed, then scraped with a razor blade and subsections of the sponge were preserved in 80% ethanol and in DNA buffer (Seutin, White & Boag, 1991) and stored on ice until returning to the laboratory. Water samples (500 mL each, $n$ = 2–3 per sampling point) were collected in acid-cleaned glass bottles and each bottle was rinsed in the river water three times before collecting water for analysis. The water was collected near the surface in arbitrary locations across the width of the river. The

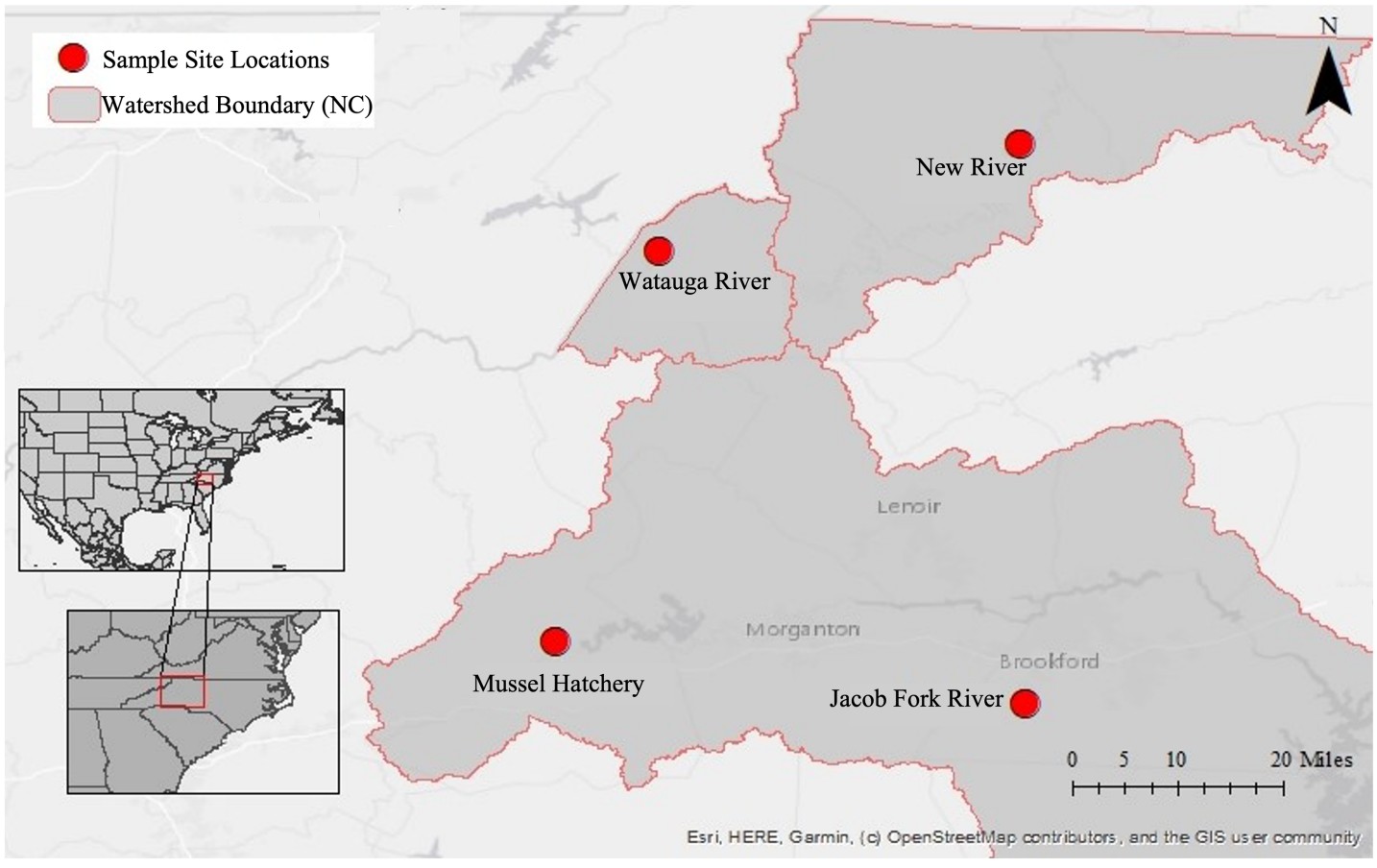

**Figure 1 Map of sampling locations in North Carolina, U.S.A.** Insets show the location within North America and the U.S.A, the second inset shows the location within the state of North Carolina. The large map has the three river basins outlined and with each sampling site marked.

mussel hatchery site contained a small pond that was used to grow mussels on nets and an indoor flow through system with pond water to maintain mussels in tanks: five sponge specimens were collected from the pond and three from the wall of indoor tanks, alongside one tank water and two pond water samples. The hypothesis that sponge and water microbiomes would differ by site was also addressed by this sampling scheme of paired sponge and water samples at each of the four sites.

To address the hypothesis of increased similarity in the sponge and water microbiomes during the growth of the sponge, at one site, the New River, we sampled four times over the growing season (July–September) and we attempted track individual sponges over time. Attempts were made to mark the rocks that sponges were collected from using flagging tape, nail polish, and spray paint, but none of these lasted in the water. Thus, as best that we can tell (based on photos of the sponge and rock), the only resampling of the same sponge individuals were the final three individuals collected from the New River in middle September, NR41, NR42, NR43, that are the same individuals as NR28, NR30, and NR29 (sampled in early September), respectively.

**Table 1 Sample collection information.** The presence of gemmules in some samples supported the identification of others without gemmules and with identical megascleres and from the same or nearby location. Site elevation is provided below coordinates and all samples were collected in 2017.

| Location | Coordinates elevation | Date (2017) | Sample ID | Spicule-based ID | Gemmules | Water sample |
|---|---|---|---|---|---|---|
| New river | 36.4158, −81.3871 | July 12 | NR16 | *Radiospongilla crateriformis* | No | NR14-15 |
| (NE) | (886 m) | July 12 | NR17 | *Radiospongilla crateriformis* | No | *n* = 2 |
| | | July 12 | NR18 | *Radiospongilla crateriformis* | No | |
| | | July 12 | NR19 | *Radiospongilla crateriformis* | No | |
| | | July 12 | NR20 | *Radiospongilla crateriformis* | No | |
| | | August 26 | NR21 | *Radiospongilla crateriformis* | No | NR26-27 |
| | | August 26 | NR22 | *Radiospongilla crateriformis* | No | *n* = 2 |
| | | August 26 | NR23 | *Radiospongilla crateriformis* | No | |
| | | August 26 | NR24 | *Radiospongilla crateriformis* | No | |
| | | September 4 | NR28 | *Radiospongilla crateriformis* | No | Not collected |
| | | September 4 | NR29 | *Radiospongilla crateriformis* | Yes | |
| | | September 4 | NR30 | *Radiospongilla crateriformis* | Yes | |
| | | September 4 | NR31 | *Trochospongilla horrida* | Yes | |
| | | September 4 | NR32 | *Radiospongilla crateriformis* | Yes | |
| | | September 17 | NR40 | *Radiospongilla crateriformis* | Yes | NR43-45 |
| | | September 17 | NR41 | *Radiospongilla crateriformis* | Yes | *n* = 3 |
| | | September 17 | NR42 | *Radiospongilla crateriformis* | Yes | |
| Watauga river | 36.2692, −81.8843 | September 4 | WR36 | *Radiospongilla crateriformis* | No | WR33-35 |
| (WA) | (985 m) | September 4 | WR37 | *Radiospongilla crateriformis* | No | *n* = 3 |
| Mussel hatchery | 35.7310, −82.0264 | September 18 | Hatch6 (pond) | *Eunapius fragilis* | No | H47-48 (pond) |
| (Hatch) | (405 m) | September 18 | Hatch7 (pond) | *Eunapius fragilis* | No | H46 (tank) |
| | | September 18 | Hatch8 (pond) | *Eunapius fragilis* | No | *n* = 1 tank, 2 pond |
| | | September 18 | Hatch9 (pond) | *Eunapius fragilis* | No | |
| | | September 18 | Hatch10 (pond) | *Eunapius fragilis* | No | |
| | | September 18 | Hatch59 (tank) | *Eunapius fragilis* | No | |
| | | September 18 | Hatch60 (tank) | *Eunapius fragilis* | Yes | |
| | | September 18 | Hatch62 (tank) | *Eunapius fragilis* | No | |
| Jacob fork | 35.6455, −81.3808 | November 21 | JF49 | *Trochospongilla* | No | JF55-57 |
| (JF) | (352 m) | November 21 | JF50 | *Trochospongilla* | No | *n* = 3 |
| | | November 21 | JF51 | *Trochospongilla* | No | |
| | | November 21 | JF52 | *Trochospongilla* | No | |
| | | November 21 | JF53 | *Trochospongilla* | No | |

In the laboratory, the ethanol in the sample tubes was replaced with fresh 80% ethanol and stored at 4 °C and sponges in DNA buffer were stored at −20 °C until DNA extraction. Water samples were filtered through a 0.22 μm polyethersulfone (PES) filter (Whatman, USA), the filter was preserved in DNA buffer and stored at −20 °C and the filtrate was frozen until used for inorganic nutrient analysis. Additional water samples were collected in 2018 for inorganic nutrient analysis and were filtered in the same process. The filtrate

from both years was used for inorganic nutrient analysis. The inorganic ions, nitrate, phosphate, sulfate, and chloride were measured using a Thermo Scientific (formerly Dionex) ion chromatograph in the Chemistry and Fermentation Sciences department at ASU equipped with an IonPac AS11-HC column set (analytical column $4 \times 250$ mm; guard column $4 \times 50$ mm) and a conductivity detector. The background conductivity was suppressed using an AERS 500 (4 mm) suppressor operated at 112 mA. The eluent was 25 mM KOH at a flow rate of 1.4 mL min$^{-1}$ under isocratic conditions. The injection volume was 20-µL and the column temp 30 °C. A standard mix at 1 ppm (Environmental Express, Charleston, SC) was used to create a standard curve for quantification of each compound. The water samples were analyzed in duplicate (technical replicates) to obtain multiple measurements. The values from the technical replicates were averaged and then sample replicates were averaged and standard deviation calculated for visualization.

Water temperature and pH were not recorded at the time of sampling in 2017. However, these data are publicly available for Cranberry Creek (36.5697, −81.327), a tributary of the New River, through the New River Conservancy (https://newriverconservancy.org/) and were used here as an indicator for changes in temperature and pH with season as opposed to absolute values of temperature and pH.

Sponge species identification was accomplished by spicule analysis. Spicule preparations were conducted by dissolving a small piece of sponge in bleach overnight in a microfuge tube. The spicules were rinsed with distilled water, then mounted on a slide to view with a compound microscope. The *Ecology and Classification of North American Freshwater Invertebrates* (*Reiswig, Frost & Ricciardi, 2010*) was used as a key for identifying the sponges. Megascleres, microscleres and gemmuloscleres (when present), were observed and used to identify the sponge.

## DNA extraction and 16S rRNA gene sequencing

DNA extractions from subsections of sponge and the whole filter were performed using the cetyltrimethylammonium bromide (CTAB) method originally designed for plant DNA extractions (*Lipp et al., 1999*). Extracted DNA was checked for concentration and quality using a Nanodrop spectrophotometer ND1000 and sent to Nova Southeastern University for sequencing on an Illumina MiSeq. Polymerase chain reaction (PCR) was performed following published protocols from the Earth Microbiome Project (http://www.earthmicrobiome.org/protocols-and-standards/). PCR reactions were run using barcoded 16S rRNA primers (515 F and 806 R: (*Caporaso et al., 2011*; *Walters et al., 2015*)). Barcoded PCR products were cleaned using AMPure beads (Beckman Coulter), and concentrations of the cleaned products were measured using a Qubit fluorometer. Cleaned products were diluted to 4 nM and then pooled in equal volumes. Sample preparation and loading onto the Illumina MiSeq followed a standard Illumina protocol, with the exception of using custom sequencing primers from the EMP protocol. Samples were sequenced using a 500 cycle V2 chemistry kit (Illumina) for 16S rRNA gene sequencing to obtain 250 bp single-end amplicons. In additional to 16S rRNA gene analysis, two marker genes of the sponge (cytochrome oxidase I (COI) and 18S rRNA) were amplified and archived (Supplemental Information), but due to poor representation of the freshwater sponge

genes in the databases, these sequences were not useful to identify the sponges in this study.

## Data analysis

The sampling map was created using ESRI ArcMap v10.8.1 Online (2024). Raw FastQ files were analyzed with the Dada2 pipeline (*Callahan et al., 2016*) within the R statistical program (*R Core Team, 2021*). Dada2 was used to trim, filter, and deduce representative amplicon sequence variants (ASVs). Within the Dada2 pipeline, reads were trimmed to 150 (forward) and 180 (reverse), with average quality scores between 20 and 30 for the remaining reads and were denoised and filtered for chimeric sequences and singletons (one read for an ASV) in Dada2. The SILVA database v138 was used as a reference ribosomal gene database. The Silva SSU database was used for taxonomic classification (v138; *Quast et al., 2013*). Phyloseq (v1.36.0; *McMurdie & Holmes, 2013*) and ggplot2 (*Wickham, 2016*) were used for downstream analysis and visualization.

Alpha and beta diversity analysis was performed with the vegan package (*Oksanen et al., 2022*) in R. Alpha diversity was compared across sample type and location with analysis of variance (ANOVA) following tests for assumptions of ANOVA and followed by Tukey's Honestly Significant difference *post-hoc* tests for all locations except for Watauga river sponges. Beta diversity analyses used relative proportions of reads for non-metric multidimensional scaling (nMDS, Phyloseq 'ordinate' function) and comparison of taxa across samples. Ordination by nMDS used Bray-Curtis or weighted Unifrac (*Lozupone & Knight, 2005*) distance matrices. Differences in composition of taxa across sample groups (*e.g.*, sponge *vs.* water samples) was deduced using a permutational analysis of variance (PERMANOVA) with the Adonis2 function in the vegan package in R. Identification of ASVs of interest between sponge and water samples (*i.e.*, taxa present or present in high proportion in one but not the other sample type) were discovered by (1) heatmap visualization of taxa abundance in each sample type, and (2) the vegan similarity percentage analysis (SIMPER) function in R comparing sample type.

An overview of the sponge and water microbiomes was conducted for all samples (*i.e.*, taxonomic profile and nMDS visualization) and then a more in-depth analysis across month was performed for the New River samples. These included visualization by ordination using nMDS (using Bray-Curtis and Unifrac distance matrices and the Phyloseq 'ordinate' function), heatmap visualization ('plot_heatmap' function from Phyloseq). PERMANOVA (Adonis2 function) was used to test for differences in the composition of symbiont taxa by month.

Differences in temperature over time at the New River site were assessed with a Kruskal-Wallis rank sum test in R (*Hollander & Wolfe, 1973*). Differences in inorganic nutrient concentrations between sites or over time were not conducted due to limited replicates.

## Data availability

The sequence data are available at NCBI in BioProject PRJNA988097 (16S rRNA gene microbiome sequences), BioProject PRJNA975882 (sponge target loci sequences), and

accession PP930792 for the 16S rRNA gene clone. The sequences are also available as Supplemental Files (sponge CO1, 18S rRNA and ITS, and one 16S rRNA gene sequence clone) and at the OSF project site for the ASV sequences (https://osf.io/rpbz7/). The raw environmental data as well as the R script and output files from Dada2 processing are available online through OSF (https://osf.io/rpbz7/).

## RESULTS

### Sponge identification

Morphological identification using spicules indicated a total of three species. All sponges from the Watauga River and the New River, except for one individual in the New River, were identified as *Radiospongilla crateriformis* (*Potts, 1882*). The megascleres were oxeas of one size class with subtle spines (Figs. S1A–S1C, S1G). The gemmuloscleres, when present (Table 1), were birotula with curved spines on each rotule and several spines at the terminal ends of the rotule shaft (Figs. S1A, S1B). For individuals without gemmuloscleres (Table 1), the high similarity in megascleres suggested that these are likely the same species. One individual of the genus *Trochospongilla* (Fig. S1I) was also observed at the New River site, with the distinctive spiny oxeas, identical to the oxea megascleres of *Trochospongilla horrida* (*Weltner, 1893*) identified from the Jacob Fork River (Fig. S1D). The *Trochospongilla* from the New River did not have gemmuloscleres and thus we refer to it as *Trochospongilla* sp., although it is likely *T. horrida* as we observed only *T. horrida* in the region. The distinctive gemmuloscleres of *T. horrida* from Jacob Fork were smooth and similar sized birotules (Fig. S1D). *Trochospongilla horrida* at Jacob Fork was present as small smooth individuals (not shown) and with a convoluted phenotype (Fig. S1L) and both tan color and green suggestive of algal symbionts. The only species observed at Jacob Fork was *T. horrida*. At the mussel hatchery, the only species observed was *Eunapius fragilis* (*Leidy, 1851*) (Figs. S1K, S1J). The hatchery individuals all had smooth and stout oxeas as megascleres and two spiny strongyle gemmuloscleres (Figs. S1E, S1F). The COI genes were nearly identical across all individuals and both the COI and 18S rRNA sequences yielded equivocal hits to multiple other species in the NCBI database (Tables S1, S2). Thus, the genetic information was not used any further and we relied on the spicule-based identifications.

### Microbial community composition in sponges and river water

For high throughput amplicon sequencing, a total of 3,287,273 reads (average of 67,087 reads per sample) were obtained from all samples. Filtering low quality and chimeric reads resulted in 1,262,913 total reads (average of 25,773 per sample) that comprised 1,359 amplicon sequence variants (ASVs).

The alpha diversity of sponge samples was higher on average than for water samples (Fig. S2, ANOVA, $F_{7,41} = 16.84$, $p < 0.001$, Shannon diversity; $F_{7,41} = 12.32$, $p < 0.001$, Inverse Simpson diversity). *Post-hoc* tests did not yield significant differences in either diversity metric for sponge *vs.* water samples at a given site (Tukey HSD, adjusted $p > 0.05$). However, *post-hoc* tests (Tukey HSD) did yield significant differences in both diversity metrics between sites for sponge samples, with the hatchery sponges having higher

diversity than Jacob Fork (adjusted $p < 0.001$ Shannon and Inverse Simpson) and New River (adjusted $p < 0.001$ Shannon and Inverse Simpson), New River sponges were higher in diversity compared to Jacob Fork sponges for Shannon diversity but not for Inverse Simpson (adjusted $p = 0.02$ Shannon, $p = 0.26$ Inverse Simpson). Alpha diversity of water samples followed the same pattern as the hatchery water samples, having a higher diversity compared to Jacob Fork (adjusted $p < 0.001$) and New River (adjusted $p < 0.001$). The New River water samples had a higher Shannon but not Inverse Simpson in diversity compared to Jacob Fork water samples (adjusted $p = 0.02$ Shannon, $p = 0.87$ Inverse Simpson).

For ASVs over 0.01% abundance, these were classified to 13 bacterial phyla. The phyla were dominated by Actinobacteria, Bacteroidota, Verrucomicrobia, Pseudomonodota, and to a lesser extent, Spirochaeta, Patescibacteria, and Chloroflexi. At the class level of taxonomy, there were also only a few taxonomic groups that comprised the most abundant ASVs particularly in the sponge samples (Fig. 2). The predominant classes in sponge samples relative to water included Verrucomicrobiae (phylum Verrucomicrobia), Parcubacteria (Patescibacteria), Leptospirae (Spirochaetota) for the Jacob Fork location, Kapabacteria (Bacteroidetes/Chlorobi group) for the hatchery location, and Alphaproteobacteria (Pseudomonodota) and unclassified ASVs across sponge samples (Fig. 2). There was a higher proportion of Actinobacteria (Actinobacteriota) and Bacteroidota (Fig. 2) in water sample communities, and similar relative abundance of Gammaproteobacteria (Pseudomonodota) and most of the low abundance classes (Fig. 2).

## Differences in microbiome composition across sample type and site

Ordination by both ecological (Bray-Curtis) and phylogenetic (weighted Unifrac) distances yielded separation in nMDS space between sponge and water microbiomes among sites (Figs. 3A, 3B). The difference in sample type (sponge *vs.* water) was supported by PERMANOVA ($F_{1,48} = 3.1$, $R^2 = 0.06$, $p = 0.003$, Bray-Curtis; $F_{1,48} = 4.38$, $R^2 = 0.08$, $p = 0.003$, Unifrac). There was also a significant effect of site for sponges (PERMANOVA, $F_{3,32} = 15.7$, $R^2 = 0.62$, $p = 0.001$,) and for water samples (PERMANOVA, $F_{3,15} = 16.2$, $R^2 = 0.8$, $p = 0.001$) (Figs. 3A, 3B). We also note that the sites tended to be species-specific and were sampled at different times of year, preventing us from testing for an effect of species or time in the microbiome composition across all of our sites. SIMPER analysis yielded 360 ASVs that explained ~80% of variation between sponge and water samples. Of the 360 ASVs, 17 ASVs explained ~30% of variation between sponge and water samples and these 17 were examined further for their taxonomic identification (four taxonomic classes: Bacteroidia, Actinobacteria, Gammaproteobacteria, Verrucomicrobiae, Table 2).

## Seasonal variability of the microbiome at the New River site

Additional nMDS ordinations examining New River samples over the months of July, August, and September also support separation by sample type and by month based on Bray-Curtis and Unifrac distance matrices (Figs. 4A, 4B). PERMANOVA supported a significant effect of sample type for New River samples (PERMANOVA, $F_{1,23} = 4.75$, $R^2 = 0.18$, $p = 0.002$ (Bray-Curtis), $F_{1,23} = 10.9$, $R^2 = 0.33$, $p = 0.001$ (Unifrac distance)) (Figs. 4A, 4B). PERMANOVA also supported a significant effect of month for New River

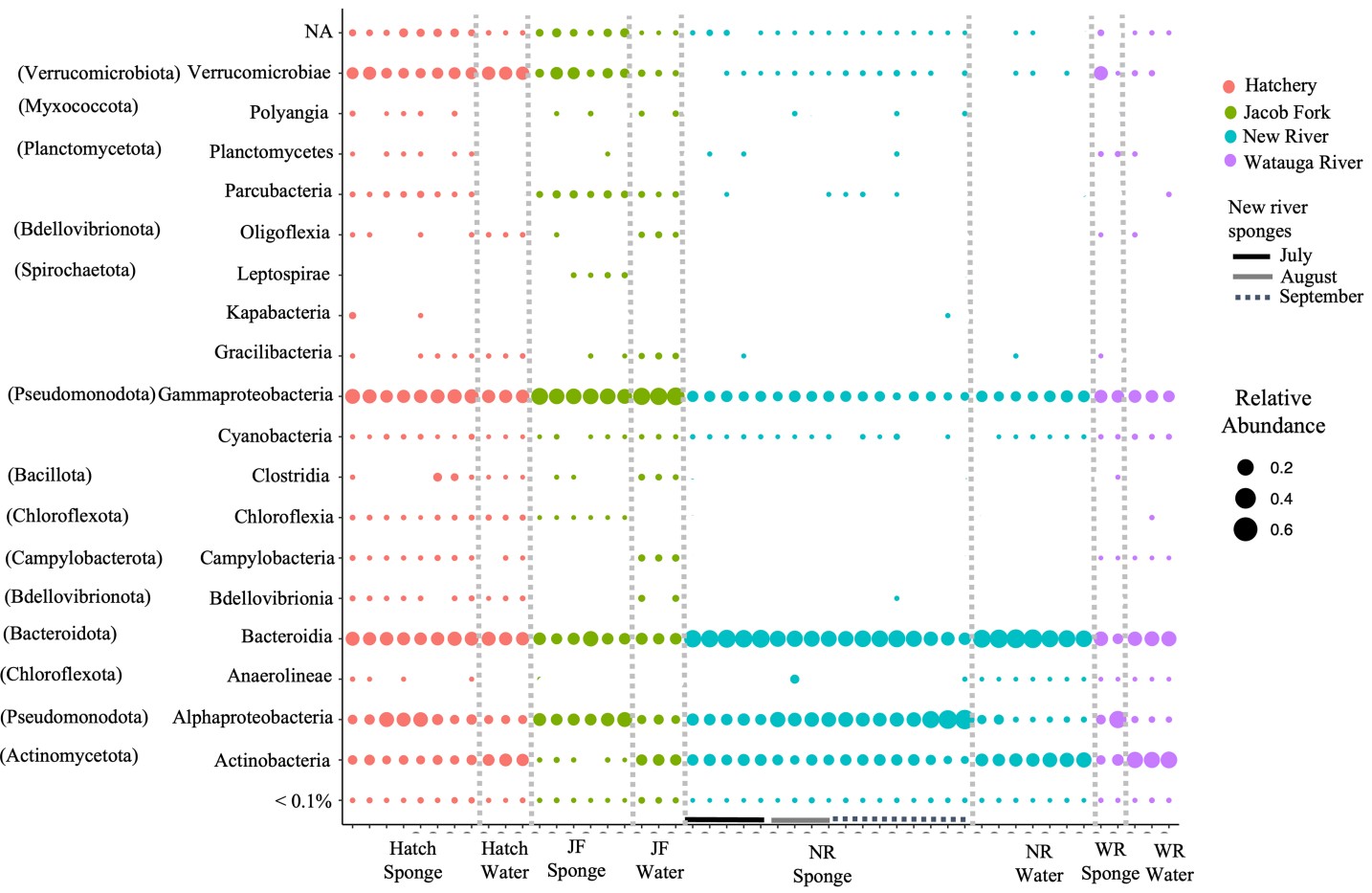

**Figure 2** **Taxonomic overview of ASVs in freshwater sponge and water samples. Relative abundance of ASVs is shown for taxonomic class.** Phylum provided in parentheses. Colors correspond to location and bars mark sponge samples and month for New River sponges. Locations = Hatchery (Hatch), Jacob Fork River (JF), New River (NR), and Watauga River (WR). NA = not classified and <0.1% includes all taxa below 0.1% relative abundance.

samples (PERMANOVA, $F_{2,23}$ = 6.8, $R^2$ = 0.39, $p$ = 0.001 (Bray-Curtis), $F_{2,23}$ = 3.5, $R^2$ = 0.25, $p$ = 0.021 (Unifrac distance). A group of three sponge samples collected in late September were unique from the early September samples in that they contained gemmules and the sponge tissue itself was degrading at the time of collection; these three samples grouped together in both New River ordinations (Figs. 4A, 4B). Because *Radiospongilla crateriformis* accounted for all individuals except for one *Trochospongilla horrida* individual at the New River site, we could not compare species within this site. It is notable, however, that the *T. horrida* sample was separated from the *R. crateriformis* individuals on the ordinations (Figs. 4A, 4B).

At the ASV level for the New River samples, the most abundant ASVs (*i.e.*, those with the most reads), included taxa that were present in both sponge and water and some taxa present only in sponges (Fig. 5 "top 50" ASVs, Fig. S4 "top 100" ASVs). More specifically, a group of taxa present in sponges and not the water samples were only present in the August and September samples, not the July samples (Figs. 5, S4, Table 3). The groups Alphaproteobacteria and Bacteroidetes accounted for the majority of these sponge-derived

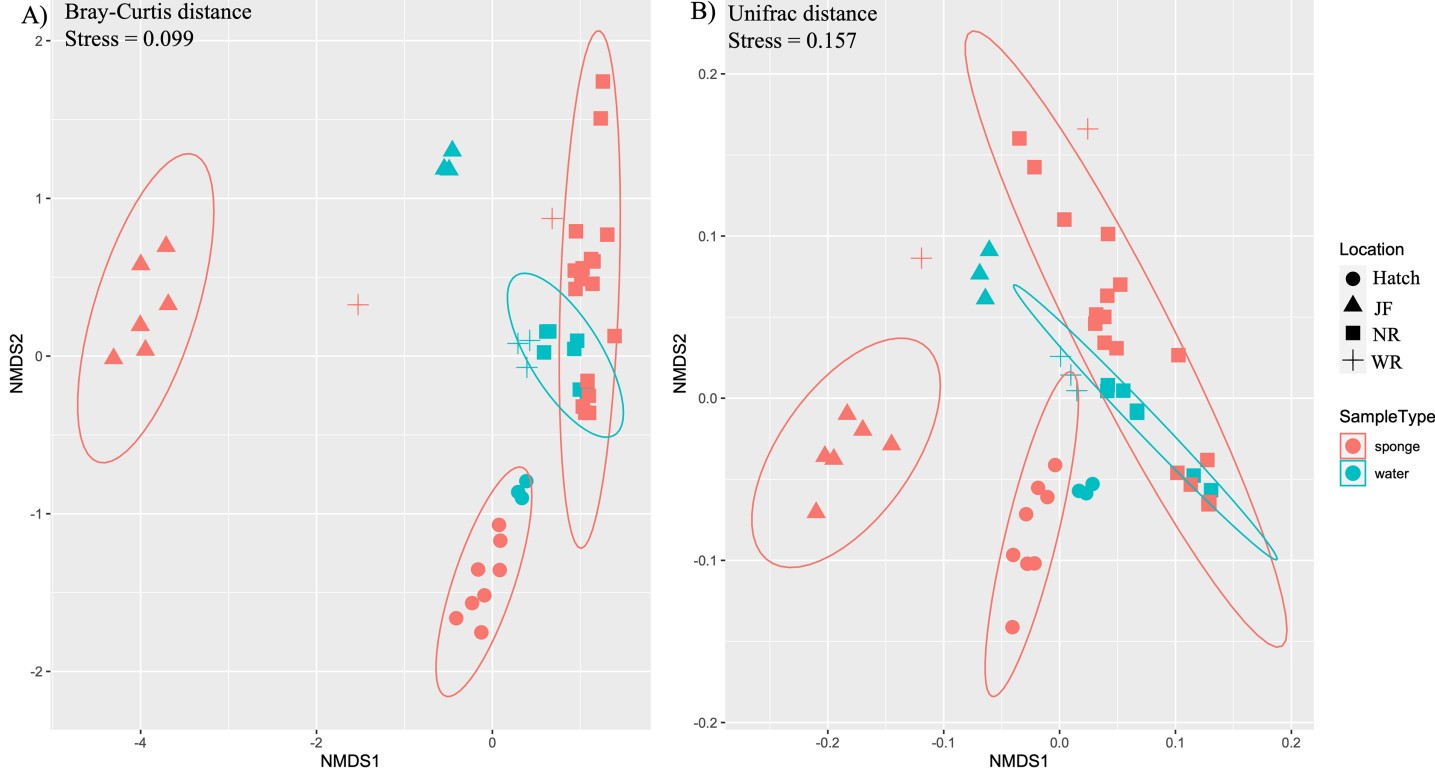

**Figure 3 Ordination of microbial communities in freshwater sponges and water based on ASVs from 16S rRNA gene profiling.** Non-metric multidimensional scaling with Bray-Curtis distance matrix (A) or Unifrac distance matrix (B) PERMANOVA analysis supports the separation of sponge and water samples ($F_{1,47}$ = 4.37, $p$ = 0.001, Bray-Curtis; $F_{1,47}$ = 3.10, $p$ = 0.002, Unifrac). Locations = Hatchery (Hatch), Jacob Fork River (JF), New River (NR), and Watauga River (WR).

taxa, with a few Gammaproteobacteria and Actinobacteria (Table 3). BLAST (*Altschul et al., 1990*) analysis with the sponge-derived Dada2 ASVs yielded similar identity (using the "top" BLAST match) to uncultured freshwater Bacteroidetes and Alphaproteobacteria groups, including bacteria from other freshwater sponges *Spongilla lacustris*, *Lubomirskia baicalensis*, and *Swartschewskia papyracea*, the latter two of which are endemic to Lake Baikal (Table 3).

## Environmental variables across sites or season

The pH of Cranberry Creek (near the New River) was stable during 2017 at 6.7 for July through October and over the years from 2013 to 2021 at ~6.5–6.7. Temperature data for Cranberry Creek did not vary significantly by month from 2013–2017 within the months of collection (July, August, September) (Kruskal-Wallis, chi-squared = 3.0526, df = 2, $p$ = 0.2173).

Inorganic nutrients were variable but generally low for chloride, nitrate, and sulfate at New River and Jacob Fork sites (Fig. S5). Phosphate was below detection in all samples. While limited replicates precluded statistical analysis, this preliminary dataset does not suggest any major differences in chloride, nitrate, and sulfate between the New and Jacob Fork sites or over time at the New River site (Fig. S5).

**Table 2  Similarity percent analysis (SIMPER) results of taxa that explain the first 30% of dissimilarity (cumsum) between all sponge and water samples.** The average relative abundance is given for each sample type. Taxa with significant *p*-values are bolded and associated taxonomy of Class and Order are shown.

| Taxa | Avg-sponge | Avg-water | Cusum | *p* | Class | Order | Family | Genus |
|---|---|---|---|---|---|---|---|---|
| **ASV1** | **0.035** | **0.068** | **0.031** | **0.001** | Actinobacteria | Frankiales | Sporichthyaceae | Candidatus Planktophila |
| **ASV2** | **0.032** | **0.068** | **0.058** | **0.001** | Bacteroidia | Cytophagales | Spirosomaceae | *Pseudarcicella* |
| ASV3 | 0.037 | 0.043 | 0.085 | 0.239 | Bacteroidia | Chitinophagales | Chitinophagaceae | *Edaphobaculum* |
| **ASV4** | **0.027** | **0.037** | **0.107** | **0.507** | Gammaproteobacteria | Burkholderiales | Burkholderiaceae | *Polynucleobacter* |
| **ASV5** | **0.019** | **0.040** | **0.130** | **0.003** | Bacteroidia | Chitinophagales | Chitinophagaceae | *Sediminibacterium* |
| ASV6 | 0.024 | 0.027 | 0.152 | 0.325 | Bacteroidia | Flavobacteriales | Flavobacteriaceae | *Flavobacterium* |
| ASV7 | 0.023 | 0.019 | 0.173 | 0.627 | Bacteroidia | Flavobacteriales | Flavobacteriaceae | *Flavobacterium* |
| **ASV8** | **0.012** | **0.043** | **0.193** | **0.001** | Actinobacteria | Frankiales | Sporichthyaceae | NA |
| ASV9 | 0.015 | 0.018 | 0.207 | 0.407 | Gammaproteobacteria | Burkholderiales | Comamonadaceae | *Limnohabitans* |
| ASV10 | 0.014 | 0.014 | 0.221 | 0.662 | Bacteroidia | Chitinophagales | Chitinophagaceae | *Sediminibacterium* |
| **ASV11** | **0.010** | **0.017** | **0.234** | **0.017** | Actinobacteria | Frankiales | Sporichthyaceae | hgcI clade |
| **ASV12** | **0.009** | **0.021** | **0.247** | **0.011** | Bacteroidia | Chitinophagales | Chitinophagaceae | *Sediminibacterium* |
| ASV13 | 0.010 | 0.013 | 0.259 | 0.224 | Verrucomicrobiae | NA | NA | NA |
| ASV14 | 0.018 | 0.000 | 0.271 | 0.924 | Bacteroidia | Chitinophagales | Chitinophagaceae | *Sediminibacterium* |
| **ASV15** | **0.007** | **0.022** | **0.282** | **0.001** | Actinobacteria | Frankiales | Sporichthyaceae | hgcI clade |
| **ASV16** | **0.005** | **0.036** | **0.294** | **0.001** | Gammaproteobacteria | Burkholderiales | Comamonadaceae | *Simplicispira* |
| ASV17 | 0.018 | 0.000 | 0.305 | 0.91 | Bacteroidia | Chitinophagales | Chitinophagaceae | *Sediminibacterium* |

## DISCUSSION

This study has described freshwater sponge microbiomes in western North Carolina, USA, and has indicated the presence of a distinct microbial community in these sponges compared to the surrounding water. These microbial communities may be transient, however, and the composition is likely influenced by both environmental and host factors. These data overall are similar to the few recent publications using high throughput sequencing on freshwater sponges (*Gladkikh et al., 2014*; *Gaikwad, Shouche & Gade, 2016*; *Laport, Pinheiro & da Costa Rachid, 2019*; *Sugden et al., 2022*; *Paix, van der Valk & de Voogd, 2024*) and builds on these studies with additional sites and sponge species, as well as the component of time at one of our sites (the New River).

Three sponge species were identified at our sites in western North Carolina, all of which are relatively common along the eastern US and have been previously documented nearby in Tennessee (*Hoff, 1943*; *Kunigelis & Copeland, 2014*). The sponge species were generally separated by site apart from one *Trochospongilla* sp. individual that was observed once at the New River site. Similar to a recent study in western Canada (*Sugden et al., 2022*), there is no or limited connection between the river and pond sites and any environmental influence on the microbial community is likely to be specific to each body of water. Differences in microbial communities were previously observed between freshwater sponge species (*Eunapius carteri* and *Corvospongilla lapidosa*) collected from two lakes in India and within the same species of sponge for *E. muelleri* and for *Eunapius fragilis*

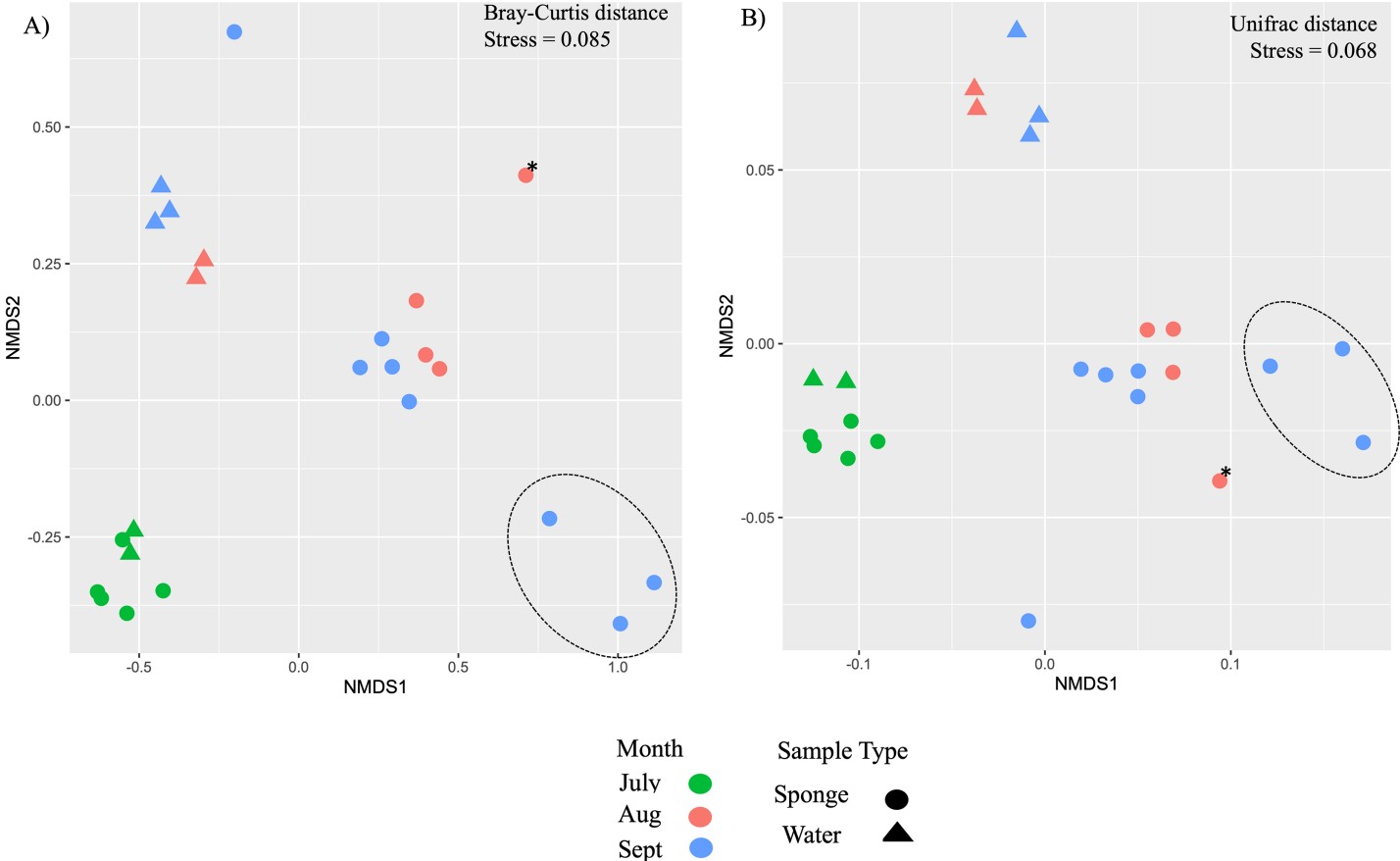

**Figure 4** **Ordination of microbial communities in freshwater sponges and water from the New River based on ASVs from 16S rRNA gene profiling.** Non-metric multidimensional scaling with Bray-Curtis distance matrix (A) or Unifrac distance matrix (B) PERMANOVA supports separation by sample type (sponge *vs.* water) and there was a significant interaction between sample type and month. Canonical correspondence analysis with Bray-Curtis distance matrix also supports some separation by month (C). The asterisk (*) on the plots denotes the *Trochospongilla* sp. sample and the dashed circle denotes sponge samples collected in late September and these contained gemmules.

(*Gaikwad, Shouche & Gade, 2016*; *Sugden et al., 2022*; *Clark et al., 2022*). These studies all support some influence of local water microbiome on the sponges at each site. In the present study, the clustering of sponge and water samples by site also suggest some environmental influence on the sponge microbiomes at each body of water. Notably, while the sites were roughly species specific, there was some overlap of species across sites (*i.e.*, *R. crateriformis* at the New River and Watagua River sites and *Trochospongilla* ($n = 1$) at New River and at Jacob Fork River), and while limited in size, these samples tended to still separate by site in the ordinations.

While we have limited physical and chemical properties of the water at each site from this project, the dissolved inorganic nutrient concentrations seem unlikely to differ significantly among sites, and there may be other parameters not measured here, such as particulate organic matter that could differ between sites and influence the microbiome composition. Additionally, the role of bacteria from the sediment is not known. While the

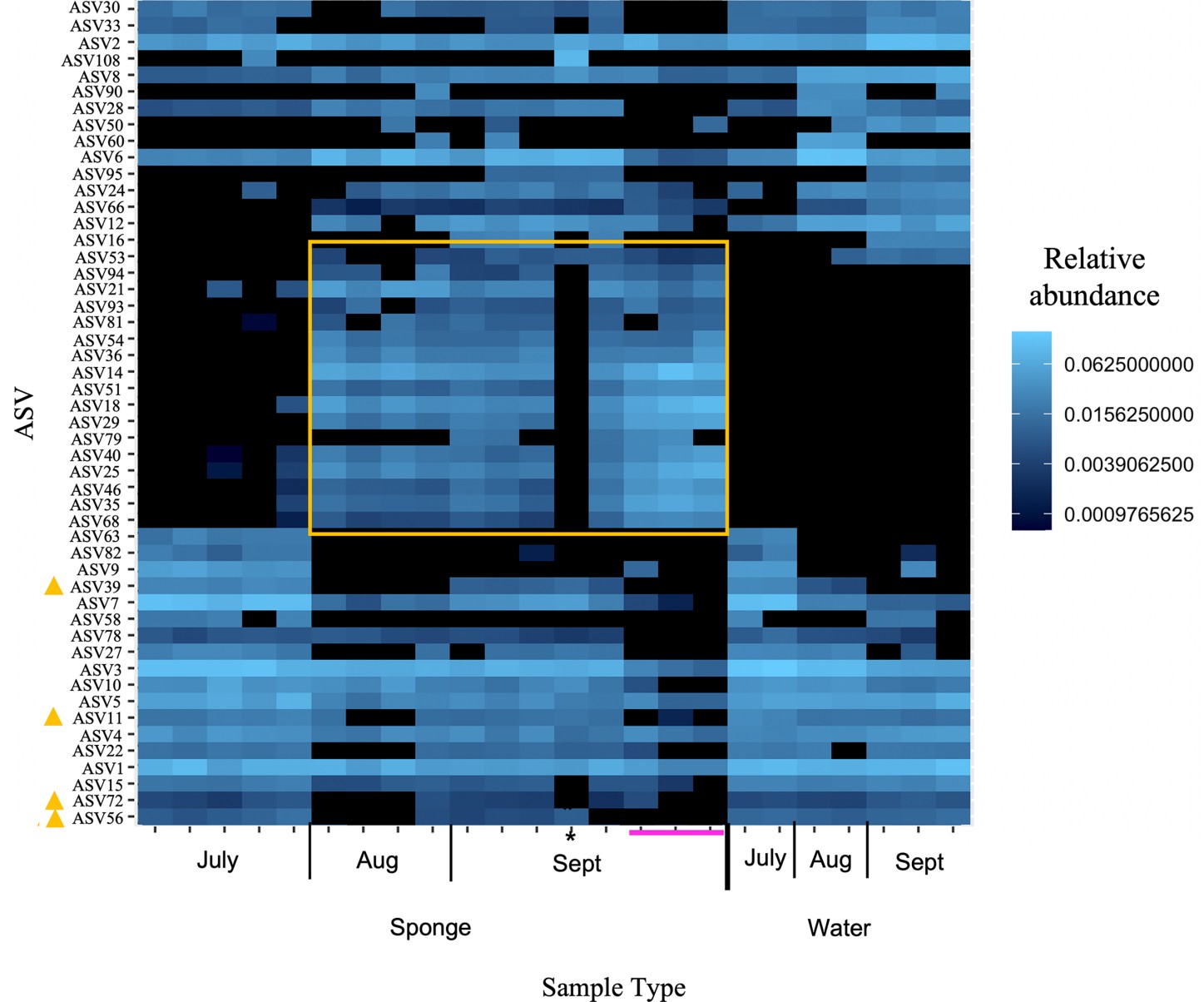

**Figure 5 Heatmap of relative abundance of "top 50" ASVs in freshwater sponge and water samples from the New River based on 16S rRNA gene profiling.** Orange box denotes taxa present only in sponges and not in corresponding water samples. Yellow triangles mark taxa present in water and sponges, but only sponges in early September. Pink line marks sponges that contained gemmules. An asterisk (*) indicates the one individual of *Trochospongilla*.

sediment microbial community is an important factor for future work, preliminary analysis in our lab of sediment microbial communities suggests low biomass in the sediment and a different community profile from the sponges (C. Fiore, 2023, unpublished data) and a recent study indicated limited overlap between freshwater sponges and nearby biofilm microbial communities (*Sugden et al., 2022*). Taken together, the water seems to be a more important source of microbes for the sponges than the benthic substrate. Further environmental data are needed to determine physical and chemical factors that may

**Table 3 Taxonomic information for ASV taxa highlighted in the heatmap comparison of sponge and water samples from the New River (Fig. 5).**

| Taxa | Phylum | Class | Order | Family | Genus |
|------|--------|-------|-------|--------|-------|
| ASV16 | Pseudomonodota | Gammaproteobacteria | Burkholderiales | Comamonadaceae | Simplicispira |
| ASV53 | Bacteroidota | Bacteroidia | Flavobacteriales | Crocinitomicaceae | Fluviicola |
| ASV94 | Pseudomonodota | Alphaproteobacteria | Rhodospirillales | NA | NA |
| ASV21 | Pseudomonodota | Gammaproteobacteria | Burkholderiales | Methylophilaceae | NA |
| ASV93 | Pseudomonodota | Alphaproteobacteria | Rhodospirillales | NA | NA |
| ASV81 | Pseudomonodota | Alphaproteobacteria | Rhodospirillales | NA | NA |
| ASV54 | Pseudomonodota | Alphaproteobacteria | Elsterales | Elsteraceae | NA |
| ASV36 | Pseudomonodota | Alphaproteobacteria | Elsterales | Elsteraceae | NA |
| ASV14 | Bacteroidota | Bacteroidia | Chitinophagales | Chitinophagaceae | Sediminibacterium |
| ASV51 | Pseudomonodota | Alphaproteobacteria | Rhodospirillales | Terasakiellaceae | NA |
| ASV18 | Pseudomonodota | Alphaproteobacteria | Rhodospirillales | Terasakiellaceae | NA |
| ASV29 | Pseudomonodota | Alphaproteobacteria | Rhodospirillales | Terasakiellaceae | NA |
| ASV79 | Pseudomonodota | Alphaproteobacteria | Rhodospirillales | Terasakiellaceae | NA |
| ASV40 | Pseudomonodota | Alphaproteobacteria | Rhodospirillales | Terasakiellaceae | NA |
| ASV25 | Pseudomonodota | Alphaproteobacteria | Rhodospirillales | Terasakiellaceae | NA |
| ASV46 | Pseudomonodota | Alphaproteobacteria | Rhodospirillales | Terasakiellaceae | NA |
| ASV35 | Pseudomonodota | Alphaproteobacteria | Rhodospirillales | Terasakiellaceae | NA |
| ASV68 | Pseudomonodota | Alphaproteobacteria | Rhodospirillales | Terasakiellaceae | NA |
| ASV39 | Bacteroidota | Bacteroidia | Flavobacteriales | Flavobacteriaceae | Flavobacterium |
| ASV11 | Actinobacteriota | Actinobacteria | Frankiales | Sporichthyaceae | hgcI clade |
| ASV72 | Bacteroidota | Bacteroidia | Cytophagales | Spirosomaceae | Pseudarcicella |
| ASV56 | Actinobacteriota | Actinobacteria | Frankiales | Sporichthyaceae | NA |
| ASV67 | Bacteroidota | Cytophagia | Cytophagales | Flexibacteraceae | Flexibacter |
| ASV152 | Pseudomonodota | Alphaproteobacteria | Rhodospirillales | Novispirillaceae | Marispirillum |
| ASV126 | Pseudomonodota | Alphaproteobacteria | Hyphomicrobiales | Blastochloridaceae | Blastochoris |

influence the sponge distribution, life cycle, and their microbial communities at these different sites.

Samples collected over several months from the New River provide insight into the seasonal environmental influence on sponge microbial communities. There is an additional factor of life cycle changes in the sponge host over time, however, and we cannot yet untangle the influences of sponge developmental changes on the sponge microbiome with environmental changes such as light, temperature, and potential changes in POC in the water. In the year that these sponges were collected, sponges were observed to form gemmules in late September simultaneous with degradation of the host tissue. There was a clear distinction in the microbiome of the sponge and river water over this time, including between the same individuals sampled in early and middle of September. The observed changes were the opposite of our hypothesis and the similarity between sponges and water in July suggests that when sponges first emerge for the season they are colonized by local

environmental microbes. We propose that as the sponge grows there are selective processes that the sponge host employs to structure the microbial community over time as evidenced by the separation of the sponge microbial community from that of the water in later summer and fall months. Similarly, a recent study demonstrated that the microbiome composition shifts with developmental changes of sponges in the laboratory for *Spongilla lacustris* (*Paix, van der Valk & de Voogd, 2024*). However, there may also be important ecological forces of microbe-microbe interactions or environmental factors (*e.g.*, temperature, dissolved nutrients, water microbial community composition) that drive changes in the sponge microbial community over time, with little influence from the host. Lastly, in the ordinations in the present study, the one *Trochospongilla* individual appeared to separate from the group of *R. crateriformis* sampled at the same time at the New River, and while preliminary, it would also suggest some host influence on the microbiome. Overall, these findings support the hypothesis that there is a mix of environmental and sponge host factors that influence the microbial symbiont assemblage (*e.g.*, *Sugden et al., 2022*; *Paix, van der Valk & de Voogd, 2024*). This phenomenon is well-documented in marine sponges (*e.g.*, *Easson & Thacker, 2014*; *Thomas et al., 2016*) and corals (*e.g.*, *Marchioro et al., 2020*), yet we are in the early stages of understanding such factors with freshwater sponges.

The dominant bacterial phyla found in freshwater sponges here include many of the same bacterial phyla or classes as previous studies on freshwater sponges from other locations around the world, particularly, a predominance of Alphaproteobacteria, Bacteroidetes, and Actinobacteria (*Kaluzhnaya, Krivich & Itskovich, 2012*; *Gladkikh et al., 2014*; *Gaikwad, Shouche & Gade, 2016*; *Laport, Pinheiro & da Costa Rachid, 2019*; *Sugden et al., 2022*; *Clark et al., 2022*). The relatively high abundance of the Verrucomicrobia is similar to the microbial communities of the Brazilian sponges of the genus *Tubella* (synonymous with *Trochospongilla pennsylvanica*), whereas this phylum was present but less abundant in sponges in Lake Baikal (*Gladkikh et al., 2014*), sponges in lakes in India (*Gaikwad, Shouche & Gade, 2016*), and in *Ephydatia muelleri* from rivers in Canada (*Sugden et al., 2022*). At the class level of microbial symbionts, however, differences between sponges in the present study and those of *Tubella* in Brazil included the high prevalence of methanotrophs in the *Tubella* sponges (*e.g.*, *Methylosinus*, *Methlylocaldum*). These differences may reflect local environmental differences that influence the sponge microbiome as some methanotrophs and other chemoautotrophs were found in the water in that study (*Laport, Pinheiro & da Costa Rachid, 2019*).

Specifically, Alphaproteobacteria were a dominant group of sponge symbionts in the most recent freshwater sponge microbiome sequencing studies (*Gaikwad, Shouche & Gade, 2016*; *Laport, Pinheiro & da Costa Rachid, 2019*; *Sugden et al., 2022*) and in the present study. The Alphaproteobacteria were more abundant in sponges compared to water while Betaproteobacteria showed the opposite trend in the *Tubella* study (*Laport, Pinheiro & da Costa Rachid, 2019*), and the relative proportions varied across river sites for the paired sponge and water samples from *E. muelleri* (*Sugden et al., 2022*). We observed a similar higher relative abundance of Alphaproteobacteria in our North Carolina

freshwater sponges although Gammaproteobacteria was the abundant counterpart in the water samples.

Taxa found to be prevalent in sponges and absent or undetectable in the water belonged primarily to the Bacteroidetes and Alphaproteobacteria and to a lesser extent, the Betaproteobacteria. The Bacteroidetes group includes a range of common environmental bacteria and this group is well documented as symbionts in vertebrates (*Thomas et al., 2011*) and invertebrates including marine (*Thomas et al., 2016*) and other freshwater sponges (*Kaluzhnaya, Krivich & Itskovich, 2012*; *Gladkikh et al., 2014*; *Gaikwad, Shouche & Gade, 2016*; *Laport, Pinheiro & da Costa Rachid, 2019*; *Hustus et al., 2023*). The recovered ASV sequences in the present study were similar to isolates of the *Flexibacter* genus, commonly found in aquatic and soil habitats and some species are known pathogens of fish (*Wakabayashi, 1986*), as well as *Sphingobacterium* and *Sediminibacterium* genera. Within the Alphaproteobacteria, taxa within the family Terasakiellaceae were highlighted as present in the New River sponges but not the water in the present study. There may be some low abundance of these taxa within the surrounding water, but this group was also described as one that is vertically transmitted within gemmules for *Spongilla lacustris* (*Paix, van der Valk & de Voogd, 2024*) and is therefore of interest for future work in this system. We highlight these groups, along with taxa within *Marispirillum* (Alphaproteobacteria, order Rhodospirillales) and *Blastochoris* (Alphaproteobacteria, order Hyphomicrobiales (synonomous with Rhizobiales)) observed in the present study as they are known for their abilities to break down diverse complex molecules such as crude oil (*Lai et al., 2009*) and chitin (*Podell et al., 2019*; *Karimi et al., 2019*), the latter of which is a component of the sponge skeleton (*Ehrlich et al., 2013*). These taxonomic groups have been observed in multiple freshwater sponge microbiome studies, including the few from North America (*Sugden et al., 2022*; *Hustus et al., 2023*). Additionally, the genomic potential and/or ability for sponge microbes to degrade complex compounds has been observed previously as characteristic of both marine and freshwater sponges (*Robbins et al., 2021*; *Sugden et al., 2022*; reviewed in *Lo Giudice & Rizzo (2024)*). Thus, these data provide further support for the presence of symbiotic bacteria within freshwater sponges that may utilize components of sponge tissue directly and/or metabolites released by the sponge or other microbes.

Strikingly, early Sanger sequencing from pilot work for this project yielded a 16S rRNA gene with high similarity to bacterial clones from the freshwater sponge *Lubomirskia baicalensis* (*Kaluzhnaya, Krivich & Itskovich, 2012*; *Kaluzhnaya & Itskovich, 2015*) that were in turn most similar to uncultured *Sediminibacterium* in the NCBI database. Similar results with high similarity to bacterial symbionts of Lake Baikal sponges and *Sediminibacterium* specifically, were also described recently for bacteria associated with *E. muelleri* (*Sugden et al., 2022*). Taxonomic groups of *Flavobacterium* and Chitinophagales also overlapped between our sponges and *E. muelleri* (*Sugden et al., 2022*; *Hustus et al., 2023*). Earlier culture-based work also suggested subtle year-to-year variability in *E. fluvatilis*-associated *Pseudomonas gac*A genes but that there was a distinct

genetic profile in the sponges compared to the water and there were persistent taxa present in all individuals (*Keller-Costa et al., 2014*). Taken together, we suggest that there were one or more ancient lineages of sponge associated taxa associated with freshwater sponges prior to diversification across the globe. In marine sponges, global network analysis of symbiont taxa revealed further support for some shared symbiont taxa across all sponges, as well as host sponge species specificity and high variability in the community due to environmental influence (*Thomas et al., 2016*). Further microbial profiling and experimental work on freshwater sponges with additional species and sites are needed to determine if there is similar specificity and shared lineages of microbial symbionts in freshwater sponges.

Overall, we observed distinct microbial community composition between freshwater sponges and the river water. The unique profile of freshwater sponge microbiome from that of the surrounding water was shown early on with culture and culture-independent (PCR-DGGE) work with *E. fluviatilis* (*Costa et al., 2013*) and has been supported by subsequent studies (*e.g.*, *Keller-Costa et al., 2014*; *Gladkikh et al., 2014*). The monthly sampling at one site revealed that sponges are likely colonized by pelagic bacteria in the water when they first emerge for the season and then through host and/or environmental selection, the community within the sponge changes in comparison to that of the water and appears to shift again (but still distinct from the water) as the sponge tissue degrades. This unique life cycle common to many freshwater sponges (and few marine sponges), presents an interesting challenge to any host specificity with their symbionts; either symbionts would need to be "packaged" in the gemmules or the host would need to regularly select for the particular symbiont(s) from the environment (*e.g.*, such as the "winnowing" process by the bobtail squid; *Nyholm & McFall-Ngai, 2004*). While recent work has demonstrated vertical transmission in freshwater sponges (*Paix, van der Valk & de Voogd, 2024*), we hypothesize that in *Radiospongilla crateriformis*, any vertically transmitted symbionts are a small proportion of the overall community. This is the first characterization of microbiomes of freshwater sponges in western North Carolina, and as far as we know, for two of the three sponge species (*Radiospongilla crateriformis* and *Trochospongilla horrida*). Further work is needed to better assess species-specificity in these microbiomes. Additional monitoring of more freshwater sponge species over time would also add to our understanding of this reservoir of microbial diversity (*e.g.*, reviewed in *Lo Giudice & Rizzo (2024)*) as well as processes and factors that shape sponge microbial symbiont communities.

## ACKNOWLEDGEMENTS

We are greatly appreciative of Dr. Carol Babyak for help with the inorganic nutrient analysis as well as to undergraduate researchers David Corcoran, Victoria Skelly, Joel Smith, Christina Strobel, Sarah Barnard, and Lauren Hagner (SUNY Cortland) for assistance in the field, laboratory, and/or data analysis. We also thank Dr. Jose Lopez at Nova Southeastern University for the use of laboratory space and sequencing on the

MiSeq. Lastly, we thank Brooke Stevens, whose work on freshwater sponge identification was critical for identification of samples collected in the present study.

### Funding

This work was supported by the Appalachian State University Office of Student Research with awards to Allison M. Griggs Mathis, Taylor A. Strope, and Jaqueline G. Keleher. The funders had no role in study design, data collection and analysis, decision to publish, or preparation of the manuscript.

### Grant Disclosures

The following grant information was disclosed by the authors:
Appalachian State University Office of Student Research.

### Competing Interests

The authors declare that they have no competing interests.

### Author Contributions

- Jacqueline G. Keleher conceived and designed the experiments, performed the experiments, analyzed the data, prepared figures and/or tables, authored or reviewed drafts of the article, and approved the final draft.
- Taylor A. Strope performed the experiments, analyzed the data, authored or reviewed drafts of the article, and approved the final draft.
- Noah E. Estrada performed the experiments, analyzed the data, authored or reviewed drafts of the article, and approved the final draft.
- Allison M. Griggs Mathis performed the experiments, analyzed the data, authored or reviewed drafts of the article, and approved the final draft.
- Cole G. Easson conceived and designed the experiments, performed the experiments, authored or reviewed drafts of the article, and approved the final draft.
- Cara Fiore conceived and designed the experiments, performed the experiments, analyzed the data, prepared figures and/or tables, authored or reviewed drafts of the article, and approved the final draft.

### Field Study Permissions

The following information was supplied relating to field study approvals (*i.e.*, approving body and any reference numbers):

New River State Park.

### DNA Deposition

The following information was supplied regarding the deposition of DNA sequences:

The sponge targeted loci sequences are available in the Supplemental Files and at NCBI: PRJNA975882. The Illumina 16S rRNA gene sequences are available at NCBI: PRJNA988097.

## Data Availability

The environmental data, R scripts, and ASV file with sequences are available at OSF: Fiore, Cara. 2024. "FWsponge 16S rRNA 2016–2017." OSF. December 7. doi:10.17605/OSF.IO/RPBZ7.

The COX1 sequences are available at NCBI: PP853474–PP853481. The 18S and ITS sequences are available at NCBI: PP853638–PP853645. The 16S clone is available at NCBI: PP930792.

## Supplemental Information

Supplemental information for this article can be found online at http://dx.doi.org/10.7717/peerj.18807#supplemental-information.

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
