# Peer review of "Freshwater sponges in the southeastern U.S. harbor unique microbiomes that are influenced by host and environmental factors"

_PeerJ, doi:10.7717/peerj.18807_

## Round 0.1 · original submission · Major Revisions

Your manuscript has now been reviewed by four external reviewers. The reviewers think that your work is of potential interest; however, they also raised major concerns about your study. In particular, reviewer #2 has made multiple suggestions on the statistical methods used in this work.

·

Basic reporting

In general, the English language should be improved to ensure that an international audience can clearly understand your text. Some examples where the language could be improved include lines 78, 87, 135, 276 – 313, the current phrasing makes comprehension difficult. The discussion needs a revision to restructure sentences which are too long and therefor hard to understand, for example line487 – 492, 503 – 508. I suggest you have a colleague who is proficient in English and familiar with the subject matter review your manuscript, or contact a professional editing service.

Experimental design

Can the authors report the replicates of the water samples per site, maybe add them also to table 1? There is no complete overview of the samples analysed in the data.

I am also wondering about the different preservation methods used (DNA buffer and ethanol), why did the authors not preserve the samples in the same way. Variation in preservation can impact bacterial community structure. Can the authors please justify their choice?

Validity of the findings

NA

Reviewer 2 ·

Basic reporting

Keleher et al. describe the host-associated microbiome in three freshwater sponge species collected across four different sampling sites in western North Carolina. Their data set represents a valuable contribution to the small but growing pool of freshwater sponge microbiome literature and provides information from new species and geographical regions. The manuscript itself is well-written, and the Introduction is both thorough and well-referenced. I only have a few comments.

Lines 82-87. “First microbiome profiling” is a questionable expression. For example, Wilkinson et al. studied green algal symbionts in freshwater sponges as early as 1980 (https://doi.org/10.1007/BF00006488). They didn’t use low- or high-throughput sequencing, but their work could still reasonably be called microbiome profiling. Consider softening the language here to avoid presenting Gernert et al. (2005) as the “first” of anything and instead just acknowledge the range of studies that have been conducted and techniques that have been used.

Lines 97-103. The authors here discuss the dichotomy of high- and low-microbial abundance sponges and present it as an open question for freshwater sponges, but their data does not assess this question because they do not measure microbial abundances in sponge tissue. Furthermore, they cite studies where sequencing-based community profiles had higher diversity than ambient water (lines 99-100), but species diversity and absolute microbial abundance are not synonymous. I recommend not broaching this question unless there is data available to answer it; I therefore suggest removing the sentence beginning “it will be interesting to see if a parallel phenomenon….”

Line 100. On a related note, Gernert et al. (2005) did not quantify microbial abundances, only RFLP-based diversity. Please remove “abundance” from this sentence.

Lines 108-109. This sentence seems to contradict both Lines 78-81 and Lines 109-110, where the authors state a general conclusion that freshwater sponges exhibit some selection of their microbial community based on host and environmental factors. Instead of commenting on the size of the database, the authors might make a stronger case by describing how the disconnected nature of freshwater ecosystems (relative to marine ecosystems) makes it more difficult to assess host- and geography-based patterns in microbiome composition.

Lines 112-114. Please clarify that Laport et al. (2019) only report this conclusion for a single target species, not as a generality of freshwater sponges.

Lines 123-124. “Would have to” should be “must.”

Lines 133-138. As with my comment above about Lines 97-103, this sentence in the Introduction about the need for DNA and RNA sequencing implies that the authors are about to fill that knowledge gap, but they provide no RNA sequencing data, and their amplicon-based DNA sequencing does not address functional relationships. I suggest removing these sentences and keeping this paragraph focused on gemmules and symbiont transmission, which is one of the strongest components of this article; the paragraph could end more impactfully on the need to sample individuals throughout a season and during gemmule production.

Lines 147-151. I suggest removing these sentences about pilot cloning data and associated study expansion – see comments below under “Experimental Design.”

Experimental design

The laboratory methods are sound; however, I have several suggestions for how the methods are presented, how the statistical analysis should be performed, and what data the authors should include in the manuscript vs. the supplement.

Major comments

Methods should clearly support hypotheses – At the end of the Introduction, the authors lay out hypotheses that (i) sponge microbial communities will be distinct from ambient water and vary by species and site and (ii) sponge microbial communities will become more similar to ambient water throughout the growing season. I suggest that the authors remind the readers of these hypotheses throughout their Methods section by modifying the topic sentences of their paragraphs to clarify the purpose behind each analysis. For example, given their hypotheses, I am not sure why the authors needed to perform flow cytometry on water samples from different streams. This data is certainly valuable, but why did the authors collect it?

Cloning data - I understand the value of preliminary studies, as the authors describe for their one Radiospongilla crateriformis sample from 2016, but I worry that this additional sample clutters the manuscript. As the authors note in their supplement, only one of their ten clones even warranted a comment in the main manuscript. I agree that the similarity between this putative Sediminibacterium sequence, the sequences from Sugden et al. 2022, and sequences from Lake Baikal sponges raises important questions about the evolution of freshwater sponge symbionts, but these questions are equally raised by the presence of Sediminibacterium sequences in their high-throughput sequencing data (as reported in Lines 572-575). Unless the authors can make a compelling case for why the inclusion of this single clone provides more or better information than their high-throughput data, I suggest removing the preliminary cloning data from the main text, which would also help make the Introduction (remove Lines 147-150), Methods (remove Lines 226-236), and Results (remove Lines 356-361) more straightforward to the reader.

Sponge identification – neither Radiospongilla crateriformis nor Trochospongilla horrida have published genomes (or even much partial representation in the NCBI database), so the inconclusive results for the COI and 18S rRNA gene sequences (Lines 347-352) are unsurprising. Because the molecular data isn’t really used, I also suggest moving most of those Methods (currently Lines 251-272) to the supplement. Instead, in the paragraph on sponge identification, the authors could say something to the effect of ‘COI and 18S rRNA gene sequences were generated and archived for a subset of samples (see Supplemental Methods), but due to the poor representation of freshwater sponges in sequencing databases, they could not be used to identify the sponges in this study.’

Other comments - Methods

Lines 159-169. Some context for the size of the streams/rivers would be helpful – that could come in the form of stream order, flow rate at the time of sampling, or even stream width. I would also recommend including a brief description of the watersheds – are these agricultural regions or forested mountains? (Some of this appears in Lines 494-498 but could be moved up here). On a related note, at what river depth were samples collected? Was this consistent across streams?

Lines 159-163. The coordinates and elevations are shown in Table 1 and, to some extent, Figure 1; I recommend referencing Figure 1 and Table 1 here and then removing the parentheticals.

Lines 173-175. The placement of this sentence (on the month of sample collection) suggests that the timing of sampling applies to the water samples but not necessarily the sponge samples. This sentence might be more effectively incorporated around Line 160 or Line 163, in the previous paragraph and before any specific sample collection is described.

Lines 175-178. Likewise, this description of sponge sampling at the hatchery is confusing in a paragraph for which the topic sentence is water samples. I suggest moving this information before the description of water sampling or adjusting the topic sentence of this second paragraph.

Lines 209. Was the flow cytometry analysis performed on the 2017 water samples or the 2019 water samples? Why was this analysis run in multiple months for Jacob Fork when sponges were only collected for one month? Most importantly, how does this data support the research question?

Lines 249. “rRNA” should appear after 16S.

Lines 264. “a subset of samples.” How many samples from which rivers were used for molecular analysis? These samples could be, for example, annotated in Table 1.

Lines 279-280. It is quite unusual to trim forward reads more stringently than reverse reads – typically, quality declines more quickly on the reverse reads. Is this correct?

Lines 289-290. Please clarify that you ran two separate ordinations with two separate distances (the text currently implies a singular ordination based on both distances). Was this weighted or unweighted uniFrac?

Lines 293. “ASVs of interest.” What makes an ASV interesting? By using SIMPER, it appears that the authors are looking for ASVs that contribute most significantly to observed beta-diversity patterns; this could be stated explicitly.

Results

Lines 347-351. This may be a mistake on my end, but I could not locate Tables S1 or S2 in the supplementary information document. As noted above, the presentation of the COI/18S rRNA results could be streamlined given that the target species have little representation in official databases.

Lines 365-375. Instead of a series of two-way comparisons, I suggest testing for differences in alpha diversity using a (generalized) linear model with both location and sample type as predictors and then reporting the coefficients for each predictor. When describing these differences, it may also be important to remind the reader that the comparisons among locations are also cross-species comparisons.

Lines 380-387. Qualitative comparisons such as these (e.g., ‘generally higher proportion,’ ‘similar relative abundance’) can be somewhat informative. To make claims about which taxa are genuinely more abundant in sponge or water samples, I suggest running some form of differential abundance analysis or repeating the SIMPER analysis at higher taxonomic levels.

Lines 397-407. Several the claims in this paragraph may be misleading given the number of potential confounds. Species is almost perfectly nested within site (the only exception is sample NR31), and, except for the New River samples, site is nested within month, so variance cannot reliably be partitioned among those three variables with such a small sample size. Furthermore, the adonis function used in the author’s R file is order-dependent, with terms added sequentially: it assigns as much variation as possible to the first term before adding the next term, which is possibly where the 52% of variation due to species comes from. The authors would need to use adonis2 and set by = “margin” to prevent this default approach, but even then, sample size is a challenge. This challenge is even acknowledged implicitly in Figure 3, where the authors chose not to indicate different species at all. I suggest that the authors limit their claims to (i) site-based differences among streams, with the acknowledgement that these could be driven by species or other unmeasured variables; and (ii) month-based differences at the New River site only, as they do in the following section (L. 418-432). Leave month and species out of the PERMANOVA scheme because they are too confounded by site.

Lines 409-413, Lines 437-443 / Tables 2 and 3. Is it possible to provide higher-resolution taxonomic information for these ASVs? Genus- or even family-level classifications, where available, are much more informative and very achievable with amplicon data sets.

Lines 424-426. What was/were the constraining variable(s) in the CCA? I notice that Figure 4C says “Bray-Curtis” on the bottom, but CCA is conventionally defined as a constrained ordination based on Euclidean distances. The design of this analysis should be more clearly explained in the Methods (see Line 400), and the code for this analysis should be added to the Rmd file – I find the code for producing plots, but not for running the CCA, so I cannot ascertain whether this is a true CCA or a dbRDA.

Lines 423. I believe this should be a reference to Figure 4.

Lines 433-437. Why was this analysis (taxa present in sponges but not water samples) only performed for the New River sponges?

Table 1. Should the New River abbreviation be “NR” instead of “NE”? Whichever gets used, I recommend consistently throughout the manuscript.

Validity of the findings

The conclusions generally follow from the data and are well-interpreted in the context of current literature. The data are easily accessible via an online repository, though I had a few notes (see previous section) on specific lines of code that I could not find. I otherwise have only a few comments about the conclusions the authors research in their Discussion.

Lines 485-535. This section is quite repetitive, especially when it comes to highlighting among-stream variation. Specifically, the content of Lines 485-498, 513-520, and 529-535 all highlight the existence of (i) symbiont selection by the sponge host and (ii) among-stream variation in sponge microbiome composition. I suggest merging these sections together to be more cohesive, and then creating separate paragraph(s) about the taxonomy-based distinctions.

Line 506. I believe the authors mean “abundant” instead of “prevalent.”

Lines 531-532, 550-552. Sponges have been shown to feed selectively (e.g. 10.1002/lno.10287). It should therefore be acknowledged that selective filter-feeding (i.e., sponge preferring to eat certain bacterial taxa) could produce the same patterns as selection for microbial symbionts – although seemingly unlikely given the literature, it cannot be disproven with this data set and could certainly be true for a certain proportion of “sponge-associated” bacterial signatures.

Lines 579. I recommend mentioning here the composition of the sponge skeleton (e.g., chitin or other structural molecules), which would provide context for why these taxa would be found associated with the sponge.

Lines 597-602. This may be a bit of an overreach given the data. True, the existence of an extremely similar sequence across such disparate geographic regions as Lake Baikal, Vancouver Island, and North Carolina is a signature of some meaningful host-microbe association – but these three data points alone are not yet enough to claim that this putative Sediminibacterium (or other symbionts) were obligate symbionts at the time of freshwater sponge diversification. I suggest softening this language to suggest that these data provide evidence of some evolutionary association, though the phylogenetic and geographic breadth of this association require further exploration.

Lines 608. As noted for the intro – “first” is a tricky word here. I suggest keeping this more general rather than focusing on specific studies.

Reviewer 3 ·

Basic reporting

See "Additional Comments"

Experimental design

See "Additional Comments"

Validity of the findings

See "Additional Comments"

Additional comments

• Consider including Clark et al (ISME Comm, 2022, 2, 22) in the introduction. They documented specific differences between genus and species level diversity in two freshwater sponges in close proximity. Additionally, this seems particularly relevant in l513-517.
• L127: “, however…”
• L135: “freshwater”
• L142: “water”
• L143: “July,”
• L196, 263, 264, remainder of manuscript: liters should be capitalized “L”
• L346: “K, J” (space added)
• Tables S1 and S2 do not appear to be a part of the Supplementary Information
• L391-393: sentence a bit vague. Visual inspection of Figs 3A and B show similarity between NR water and sponge samples. Comment? Significant differences?
• L432: Do the authors intend to write “Figure 4 A,B,C”?
• L492: “a little”
• L497: replace “flesh out”, colloquial
• L624-625: E. fragilis, see Clark et all ISME Comm (ISME Comm, 2022, 2, 22), this was a ‘culturable’ microbiome analysis.
• The authors have done a thorough job of comparing communities at several different sampling sites between several samples and have adhered to methodological norms used in peer-reviewed literature for studies of this type. Therefore this is a welcomed addition to a slim body of work that documents microbial communities within freshwater sponges. Commentary: This being said, it is difficult to draw too many conclusions between marine v freshwater; sample site A v B; etc when using phylum populations as a comparison. Generally speaking, use of phylum (and even Class) is common to compare bacterial communities between hosts. I wonder how significant this comparison is, since that is a very wide ‘net to cast’. Example: Streptomyces and Mycobacterium are both genera within the phylum Actinomycetota, yet both harbor species with incredibly different functions (coelicolor v tuberculosis). This is a rule in the microbial world, not an exception. The study is of publishable quality given some minor changes, though if the authors have the opportunity to perform comparisons at more specific taxonomic levels, this would add much value to the study. The Reviewer understands that this may not be possible given the type of data collected in this study and that further work may be out of the scope of the current manuscript.

Reviewer 4 ·

Basic reporting

In the present study, the others explore the Freshwater sponges in the southeastern U.S. harbor unique microbiomes that are influenced by host and environmental factors. This is a very interesting paper. From my inspection of the manuscript, it could be accepted after major revisions.
My main comments are as follows:
Comment 1: I recommend that the authors add the aim of their study at the end of the introduction by highlighting importance of understanding factors influence freshwater sponge.

Experimental design

Comment 2: In the material and methods, the number of samples from water and sponges understudies is not mentioned.
Comment 3: In the material and methods, DNA buffer composition or producing company not mentioned.
Comment 4: Line 233, “The PCR product was cloned with Escherichia coli and ten clones were selected for plasmid DNA extraction and Sanger sequencing at Eurofins Genomics” Why the authors did cloning?
Comment 5: Line 265, ‘’A subset of samples was used for amplification of the sponge genes because for multiple samples there was no DNA left after use for microbiome profiling’’.
Please remove this sentence

Validity of the findings

Comment 6: Line 347: Why do the COI and 18S rRNA from the author's vision yield equivocal hits to multiple other species in the NCBI database? Please remove the experimental work related to this part
Comment 7: Line 430, explain the result within line 430-432.
Comment 8: Line 584, No evidence for symbiotic relation in the manuscript, rewrite this paragraph being careful with words.
Comment 9: discussion should be improved to express the outcome of the research

---

## Round 0.2 · Minor Revisions

There are still a few minor concerns from reviewer #2 that need to be addressed.

Reviewer 2 ·

Basic reporting

I was Reviewer 2 of the initial manuscript. I appreciate the author’s work to streamline the presentation of their story, clarify their statistical analyses by acknowledging confounds, and add a stronger sense of purpose to their methods. I think the manuscript has been tremendously improved. Although I have several additional comments below, these concerns are all minor and could be addressed quickly – typographical errors, unclear sentences, or repetitiveness that was introduced in the revisions. Please note that my line numbers below refer to the line numbers in the PDF version of the “clean” document (with no tracked changes).

Line 107. I’m fine if the authors would like to keep this paragraph focused on colonization, though the transition from the previous paragraph could be smoothed by stating the meaning of “database” explicitly here. “There is not yet enough data on freshwater sponge-associated microbiota to draw general conclusions…”

Lines 137-139. I like the additional sentence here, which gives a much stronger sense of purpose. However, I recommend a minor adjustment in tense and phrasing to increase the impact of this sentence pair: “additional DNA and RNA sequencing *will be* needed… taxonomic profiling *remains a necessary* first step…”

Line 158. With this text moved down from the Introduction section, please provide specific years for “the first year of sponge collections in this region” (i.e., 2016) and “the following year” (i.e., 2017). The reader otherwise does not have context for these expressions until later in this paragraph.

Lines 162-166. With the addition of temporal information here (i.e., months of sampling), this sentence (“sponges and corresponding water samples were collected…”) perfectly repeats the sentence at Lines 195-198 (“sponge and water collections were made…”). Please state this information only once.

Lines 170-172. This sentence describing the methods of sponge collection belongs after the site description. It could be placed as the second sentence in the paragraph that now begins “to address the hypothesis of differences in sponge and water microbiomes…”

Lines 173-189. Thanks for this additional context on your study sites – this is very helpful.

Lines 190-192. Consider blending the sentence in L.202-204 with this opening sentence of the paragraph – something like: “To address our hypotheses that sponge microbiomes be different both (i) from the ambient water and (ii) across our various study sites, we collected sponge and water samples at the same time during each sampling event at our four sites.” Then delete the sentence in L.202-204.

Line 235. I understand that the authors removed their flow cytometry analysis in response to my earlier questions about its relevance. I agree with the authors that the manuscript reads more cleanly without this data, though I’m sure the authors invested a fair amount of time and money into obtaining their flow cytometry results so I do encourage them to find a future home for this data.

Line 260. “Two marker genes.” Overall, I support this more condensed presentation, but I think the genes should still be named here (e.g., “we amplified and sequenced the cytochrome oxidase (COI) and 18S rRNA genes of the sponge, but due to…”). Also, L.261, delete “the” before “freshwater sponge genes.”

Lines 350-354. This sentence has become garbled in the revision. Perhaps start a new sentence at L352, “The New River water samples had a higher Shannon but not inverse Simpson diversity compared to Jacob Fork (adjusted p = 0.02 and 0.87, respectively).”

Lines 371-372. “across sites and between sites” = “among sites”

Lines 376-377. I appreciate this simplification of the site-based differences and acknowledgement of the confound. I would encourage the authors to explicitly state the (statistical) consequences of this experimental design: “We also note that our sites tended to be species-specific are were sampled at different times of year, preventing us from testing for species- or, with the exception of the New River site (see below), time-based differences in microbiome composition.”

Line 388. Typo. PERMANVOA = PERMANOVA.

Experimental design

My initial comments on the methods and statistical analysis have been sufficiently addressed.

Validity of the findings

Lines 437-491. The new organization of the Discussion is much stronger, though the second and third paragraphs of this new Discussion are a bit wordy/repetitive and could be revised for concision.

Lines 548-550. I see that the authors changed “symbiotic” to “sponge-associated” here in response to my initial comment. I was perhaps not clear initially, but I’m more broadly concerned with the basis for this suggestion that ancient, evolutionary sponge-microbe associations (whether symbiotic or not) predated geographic diversification. This suggestion—even though it will likely prove to be accurate—overextends the authors’ data without adequately acknowledging this overextension. Indeed, the primary basis of this conclusion is the occurrence of a single sequence, Sediminibacterium, across a handful of species in three disparate habitats. I agree with the authors that this is remarkable, but compare that to Thomas et al. (2016), cited at L. 553, who used 81 species across 20 countries to make claims of evolutionarily conserved associations. We’re simply not there yet with freshwater sponges.

If the authors want to make this suggestion, it must be presented with an appropriate acknowledgement of the uncertainty. For example: “Taken together, these disparate occurrences of a similar Sediminibacterium sequence provide increasing evidence for geographically conserved sponge-microbe associations. While we suspect that this association predates the diversification of freshwater sponges across the globe, untangling the true phylogenetic and geographic breadth of this association will require data from many more species and sites.”

Line 557. Because of the confound between species and site – which has now been sufficiently addressed in the results (thanks) – the authors have not definitively shown distinct microbial compositions “between sponge species.” Reframe this sentence to be tighter to the revised results.

---

## Round 0.3 · accepted · Accept

The authors have addressed all of the reviewers' comments. This manuscript is now ready for publication.